# TRAJECTORY GROWTH LOWER BOUNDS FOR RANDOM SPARSE DEEP RELU NETWORKS

## ABSTRACT

This paper considers the growth in the length of one-dimensional trajectories as they are passed through deep ReLU neural networks, which, among other things, is one measure of the expressivity of deep networks. We generalise existing results, providing an alternative, simpler method for lower bounding expected trajectory growth through random networks, for a more general class of weights distributions, including sparsely connected networks. We illustrate this approach by deriving bounds for sparse-Gaussian, sparse-uniform, and sparse-discrete-valued random nets. We prove that trajectory growth can remain exponential in depth with these new distributions, including their sparse variants, with the sparsity parameter appearing in the base of the exponent.

## 1 INTRODUCTION

Deep neural networks continue to set new benchmarks for machine learning accuracy across a wide range of tasks, and are the basis for many algorithms we use routinely and on a daily basis. One fundamental set of theoretical questions concerning deep networks relates to their *expressivity*. There remain different approaches to understanding and quantifying neural network expressivity. Some results take a classical approximation theory approach, focusing on the relationship between the architecture of the network and the classes of functions it can accurately approximate (Lu et al. (2017); Cybenko (1992); Hornik et al. (1989)). Another more recent approach has been to apply persistent homology to characterise expressivity (Guss & Salakhutdinov (2018)), while Poole et al. (2016) focus on global curvature, and the ability of deep networks to disentangle manifolds. Other works concentrate specifically on networks with piecewise linear activation functions, using the number of linear regions (Montufar et al. (2014)) or the volume of the boundaries between linear regions (Hanin & Rolnick (2019)) in input space. In 2017, Raghu et al. (2017) proposed trajectory length as a measure of expressivity; in particular, they consider the expected change in length of a one-dimensional trajectory as it is passed through Gaussian random neural networks (see Figure 1 for an illustration). Their primary theoretical result was that, in expectation, the length of a one-dimensional trajectory which is passed through a fully-connected, Gaussian network is *lower bounded* by a factor that is exponential with depth, but not with width.

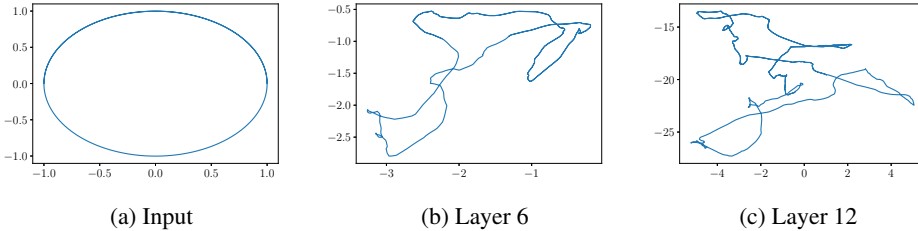

|         (a) Input         |        (b) Layer 6        |        (c) Layer 12       |

Figure 1: A circular trajectory, passed through a ReLU network with $\sigma_w = 2$. The plots show the pre-activation trajectory at different layers projected down onto 2 dimensions.

One-dimensional trajectories and their evolution through deep networks are also of interest in their own right because they constitute simple data manifolds. Firstly, we commonly assume that the real

data which we aim to correctly classify or predict with a deep network lie on one or more manifolds, and thus design a network to perform appropriately on such a manifold. Secondly, researchers are beginning to consider whether the *output* (manifolds) of generator networks could be a good model for real word data manifolds, for example, as priors for a variety of inverse problems (Manoel et al. (2017); Huang et al. (2018)). Both of these hypotheses motivate an understanding of how manifolds are acted upon by deep networks.

Our results in this paper pertain specifically to the 'trajectory length' measure of expressivity. We produce a simpler proof than in the pioneering work of Raghu et al. (2017), which also generalises their results, deriving similar lower bounds for a broader class of random deep neural networks.

Theoretical work of this nature is important because it allows for more straightforward transfer and adaptation of prior theoretical results to new contexts of interest. For example, there is a current surge in research around low-memory networks, training sparse networks, and network pruning. Sparsely connected networks have shown the capacity to retain very high test accuracy (Frankle & Carbin (2019); Han et al. (2015)), increased robustness (Ahmad & Scheinkman (2019); Aghasi et al. (2017)), with much smaller memory footprints, and less power consumption (Yang et al. (2019)). The approach we take in this work enables us to extend results from dense random networks to sparse ones. It also allows us to consider the other weight distributions of sparse-Gaussian, sparse-uniform and sparse-discrete networks (see Definitions 2 - 4).

More specifically we make the following contributions:

**Contributions:**

1. We provide an alternative, simpler method for lower bounding expected trajectory growth through random networks, for a more general class of weights distributions (Theorem 2).

2. We illustrate this approach by deriving bounds for sparse-Gaussian, sparse-uniform, and sparse-discrete random nets. We prove that trajectory growth can be exponential in depth with these distributions, with the sparsity appearing in the base of the exponential (Corollaries 1 - 3).

3. We observe that the expected length growth factor is strikingly similar across the aforementioned three distributions. This suggests a universality of the expected growth in length for iid centered distributions determined only by the variance and sparsity (Figure 3).

## 1.1 NOTATION

We consider feedforward ReLU deep neural networks. We denote a the $d$-th post-activation layer as $z^{(d)}$, and the subsequent pre-activation layer as $h^{(d)}$, such that

$$h^{(d)} = W^{(d)} z^{(d)} + b^{(d)}, \qquad z^{(d+1)} = \phi(h^{(d)}),$$

where $\phi(x) := \max(x, 0)$ is applied elementwise. We denote $x = z^{(0)}$.

We use $f_{NN}(x; \mathcal{P}, \mathcal{Q})$ to denote a random feedforward deep neural network which takes as input the vector $x$, and is parameterised by random weight matrices $W^{(d)}$ with entries sampled iid from the distribution $\mathcal{P}$, and bias vectors $b^{(d)}$ with entries drawn iid from distribution $\mathcal{Q}$.

**Definition 1.** *A **random sparse network** with sparsity parameter $\alpha$, denoted $f_{NN}(x; \alpha, \mathcal{P}, \mathcal{Q})$, is a random feedforward network in which all weights are sampled from a mixture distribution of the form*

$$w_{ij} \sim \alpha \mathcal{P} + (1 - \alpha)\delta,$$

*where $\delta$ is the delta distribution at 0, and $\mathcal{P}$ is some other distribution. In other words, weights are 0 with probability $1 - \alpha$, and sampled from $\mathcal{P}$ with probability $\alpha$. Biases are drawn iid from $\mathcal{Q}$.*

**Definition 2.** *A **sparse-Gaussian network** is a random sparse network $f_{NN}(x; \alpha, \mathcal{P}, \mathcal{Q})$, where $\mathcal{P} = \mathcal{N}(0, \sigma_w^2)$ and $\mathcal{Q} = \mathcal{N}(0, \sigma_b^2)$.*

**Definition 3.** *A **sparse-uniform network** is a random sparse network $f_{NN}(x; \alpha, \mathcal{P}, \mathcal{Q})$, where $\mathcal{P} = \mathcal{U}(-C_w, C_w)$ and $\mathcal{Q} = \mathcal{U}(-C_b, C_b)$.*

**Definition 4.** *A **sparse-discrete network** is a random sparse network $f_{NN}(x; \alpha, \mathcal{P}, \mathcal{Q})$, where $\mathcal{P}$ is a uniform distribution over a finite, discrete, symmetric set $\mathcal{W}$, with cardinality $|\mathcal{W}| = N_w$, and $\mathcal{Q}$ is a uniform distribution over a finite, discrete, symmetric set $\mathcal{B}$, with cardinality $|\mathcal{B}| = N_b$.*

For a weight matrix $W$ in a random sparse network, with $w_i$ denoting the $i^{\text{th}}$ row, we define $w_{\mathcal{P}_i}$ as the vector containing only the $\mathcal{P}$-distributed entries of $w_i$.

We define a trajectory $x(t)$ in input space as a curve between two points, say $x_0$ and $x_1$, parameterized by a scalar $t \in [0,1]$, with $x(0) = x_0$ and $x(1) = x_1$, and we define $z^{(d)}(x(t)) = z^{(d)}(t)$ to be the image of the trajectory in layer $d$ of the network. The trajectory length $l(x(t))$ is given by the standard arc length,

$$\int_t \left\| \frac{dx(t)}{dt} \right\| dt.$$

As in the work by Raghu et al. (2017), this paper considers trajectories with $x(t + dt)$ having a non-trivial component perpendicular to $x(t)$ for all $t, dt$.

Finally, we say a probability density or mass function $f_X(x)$ is even if $f_X(-x) = f_X(x)$ for all random vectors $x$ in the sample space.

## 2 EXPECTED TRAJECTORY GROWTH THROUGH RANDOM NETWORKS

Raghu et al. (2017) considered ReLU and hard-tanh Gaussian networks with the standard deviation scaled by $1/\sqrt{k}$. Their result with respect to ReLU networks is captured in the following theorem.

**Theorem 1** (Raghu et al. (2017))**.** *Let $f_{NN}(x; \mathcal{N}(0, \sigma_w^2/k), \mathcal{N}(0, \sigma_b^2))$ be a random Gaussian deep ReLU neural network with layers of width $k$, then*

$$\mathbb{E}[l(z^{(d)}(t))] \geq \mathcal{O}\left( \frac{\sigma_w \sqrt{k}}{\sqrt{k+1}} \right)^d \cdot l(x(t)),$$

*for $x(t)$ a 1-dimensional trajectory in input space.*

There are, however, other network weight distributions which may be of interest. For example, the expressivity and generative power of *sparse* networks are of particular interest in the current moment, given the current interest in low-memory and low-energy networks, training sparse networks, and network pruning (Frankle & Carbin (2019); Han et al. (2015); Yang et al. (2019)). We prove that even for sparse random networks, trajectory growth can remain exponential in depth given sufficiently large initialisation scale $\sigma_w$. Scaling $\sigma_w$ by $1/\sqrt{k}$ can yield a width-independent lower bound on this growth. Moreover, a sufficiently high sparsity fraction $(1 - \alpha)$ results in a lower bound which, instead of growing exponentially, shrinks exponentially to zero. This is captured by the following result.

**Corollary 1** (**Trajectory growth in deep sparse-Gaussian random networks**)**.** *Let $f_{NN}(x; \alpha, \mathcal{N}(0, \sigma_w^2), \mathcal{N}(0, \sigma_b^2))$ be a sparse-Gaussian, feedforward ReLU network as defined in Section 1.1, with layers of width $k$. Then*

$$\mathbb{E}[l(z^{(d)}(t))] \geq \left( \frac{\alpha \sigma_w \sqrt{k}}{\sqrt{2\pi}} \right)^d \cdot l(x(t)), \tag{1}$$

*for $x(t)$ a 1-dimensional trajectory in input space.*

Corollary 1 with $\alpha = 1$ and $\sigma_w$ replaced by $\sigma_w/\sqrt{k}$ recovers a bound which is very similar to the prior bound by Raghu et al. (2017) in Theorem 1.

Beyond Gaussian weights, we consider other distributions commonly used for initialising and analysing deep networks. Uniform distributions, for example, still constitute the default initialisations of linear network layers in both Pytorch and Tensorflow (uniform according to $\mathcal{U}(-1/\sqrt{k}, 1/\sqrt{k})$ in the case of Pytorch, and uniform according to $\mathcal{U}(-6/\sqrt{k_{in} + k_{out}}, 6/\sqrt{k_{in} + k_{out}})$ – a.k.a the Glorot/Xavier uniform initialization (Glorot & Bengio (2010)) – in the case of Tensorflow). We prove an analogous lower bound for uniformly distributed weights.

**Corollary 2** (**Trajectory growth in deep sparse-uniform random networks**). *Let* $f_{NN}(x; \alpha, \mathcal{U}(-C_w, C_w), \mathcal{U}(-C_b, C_b))$ *be a sparse-Uniform, feedforward ReLU network as defined in Section 1.1, with layers of width $k$. Then*

$$\mathbb{E}[l(z^{(d)}(t))] \geq \left( \frac{\alpha C_w \sqrt{k}}{4\sqrt{2}} \right)^d \cdot l(x(t)), \tag{2}$$

*for $x(t)$ a 1-dimensional trajectory in input space.*

Another research direction which has gathered some momentum in recent years are quantized or discrete-valued deep neural networks (Li et al. (2017); Hubara et al. (2016; 2017)), including recent work using integer valued weights (Wu et al. (2018)). This motivates consideration of discrete weight distributions, in addition to continuous ones. As an example of such, we prove a similar lower bound for networks with weights and biases uniformly sampled from finite, symmetric, discrete sets.

**Corollary 3** (**Trajectory growth in deep sparse-discrete random networks**). *Let* $f_{NN}(x; \alpha, \mathcal{P}, \mathcal{Q})$ *be a sparse-discrete random feedforward ReLU network as defined in Section 1.1, and layers of width $k$. Then*

$$\mathbb{E}[l(z^{(d)}(t))] \geq \left( \frac{\alpha \sqrt{k}}{2\sqrt{2}} \cdot \frac{\sum_{w \in \mathcal{W}} |w|}{N_w} \right)^d \cdot l(x(t)) \tag{3}$$

*for $x(t)$ a 1-dimensional trajectory in input space.*

In all cases these lower bounds show how to choose the combination of $\sigma_w$ and $\alpha$ to guarantee (or not) exponential growth in trajectory length in expectation at initialisation.

The main idea behind the derivation of these results is to consider how the length of a small piece of a trajectory (some $\|dz^{(d)}\|$) grows from one layer to the next ($\|dz^{(d+1)}\| = \|\phi(h^d(t+dt)) - \phi(h^{(d)}(t)\|$). In the context of random feedforward networks, we can consider piecewise linear activation functions as restrictions of $dh^{(d)}$ to a particular support set which is statistically dependent on $h^{(d)}$. This approach was developed by Raghu et al. (2017). The key to our proof is providing a more direct and more generally applicable way of accounting for this dependence than originally provided by Raghu et al. (2017). Specifically, our approach lets us derive the following, more general result, from which Corollaries 1, 2, and 3 follow easily.

**Theorem 2** (**Trajectory growth in deep random sparse networks**). *Let $f_{NN}(x; \alpha, \mathcal{P}, \mathcal{Q})$ be a random sparse network as defined in Section 1.1, with layers of width $k$. Let $\mathcal{P}$ and $\mathcal{Q}$ be such that the joint distribution over a vector of independent elements from both distributions is even. If $\mathbb{E}[|\boldsymbol{u}^\top \hat{w}_{\mathcal{P}_i}|] \geq M \|\boldsymbol{u}\|$ for any constant vector $\boldsymbol{u}$, for all $i$, then*

$$\mathbb{E}[l(z^{(d)}(t))] \geq \left( \frac{\alpha M \sqrt{k}}{2} \right)^d \cdot l(x(t)) \tag{4}$$

*for $x(t)$ a 1-dimensional trajectory in input space.*

**Remark.** *It is trivial to amend this result for networks where the width, distribution, and sparsity varies layer by layer, in which case the lower bound (4) is replaced by*

$$\prod_{j=i}^{d} \left( \frac{\alpha_j M_j \sqrt{k_j}}{2} \right) \cdot l(x(t))$$

*Moreover, the bounds from Theorem 2 and Corollaries 1 - 3 hold true in the 0 bias case as well.*

## 3  PROOF OF THEOREM 2

We prove Theorem 2 in three stages: i) We turn the problem into one of bounding from below the change in the length of an infinitesimal line segment; ii) we account simply and explicitly for the dependence generated by the ReLU activation; and iii) we break this dependence by taking advantage of the symmetry characterising this class of distributions. Supporting lemmas can be found in Appendix A.

*Proof.* **Stage 1:**

For the first stage of proof, we will closely follow Raghu et al. (2017). We are interested in deriving a lower bound of the form,

$$\mathbb{E}\left[\int_t \left\|\frac{dz^{(d)}(t)}{dt}\right\| dt\right] \geq C \cdot \int_t \left\|\frac{dx(t)}{dt}\right\| dt, \qquad (5)$$

for some constant $C$. As noted by Raghu et al. (2017), it suffices to instead derive a bound of the form

$$\mathbb{E}\left[\|dz^{(d)}(t)\|\right] \geq C\|dx(t)\|,$$

since integrating over $t$ yields the desired form. Our approach will be to derive a recurrence relation between $\|dz^{(d+1)}\|$ and $\|dz^{(d)}\|$, where we refrain from explicitly including the dependence of $dz$ on $t$, for notational clarity.

Next, like Raghu et al. (2017), our proof relies on the observation that

$$\begin{aligned}
dz^{(d+1)} &= \phi(W^{(d)}z^{(d)}(t+\delta t) + b^{(d)}) - \phi(W^{(d)}z^{(d)}(t) + b^{(d)}) \\
&= \phi^{(d)}(t+\delta t) - \phi^{(d)}(t) \\
&= d\phi^{(d)},
\end{aligned}$$

and that since $\phi$ is the ReLU operator, $\frac{d\phi}{dh_j^{(d)}}$ is either 0 or 1. When $z^{(d)}$ is fixed independently of $W^{(d)}$ and $b^{(d)}$, then $P(h_j^{(d)} = 0) = 0$ (see the preamble to Lemma 6 for more detail on this), and thus we need only note that $d\phi_j^{(d)} = dh_j^{(d)}$ when $h_j^{(d)} > 0$, and $d\phi_j^{(d)} = 0$ when $h_j^{(d)} < 0$. We define $\mathcal{A}^{(d)}$ to be the set of 'active nodes' in layer $d$; specifically,

$$\mathcal{A}^{(d)} := \{j : h_j^{(d)} > 0\},$$

and $I_{\mathcal{A}^{(d)}} \in \mathbb{R}^{k \times k}$ is defined as the matrix with ones on the diagonal entries indexed by set $\mathcal{A}^{(d)}$, and 0 everywhere else. We can then write

$$\begin{aligned}
\|dz^{(d+1)}\| &= \|I_{\mathcal{A}^{(d)}}(h^{(d)}(t+dt) - h^{(d)}(t))\| \\
&= \|I_{\mathcal{A}^{(d)}} W^{(d)} dz^{(d)}\|.
\end{aligned}$$

From here we will drop the weight index $(d)$ to minimise clutter in the exposition.

It is at this point where we depart from the proof strategy used by Raghu et al. (2017). The next steps in their proof depend heavily on the weight matrices in the network being Gaussian. For example, they require that a weight matrix after rotation has the same, i.i.d. distribution as the matrix before rotation. Instead, our proof can tackle a number of other, non-rotationally-invariant distributions, as well as sparse networks.

**Stage 2:**

The next stage of the proof begins by noting that after conditioning on size of the set $\mathcal{A}$,

$$\mathbb{E}[\|I_{\mathcal{A}} W dz^{(d)}\| \mid |\mathcal{A}|] = \mathbb{E}[\|\hat{W} dz^{(d)}\| \mid \hat{w}_i^\top z^{(d)} + \hat{b}_i > 0 \,\forall i, |\mathcal{A}|], \qquad (6)$$

where $\hat{W} \in \mathbb{R}^{|\mathcal{A}| \times k}$ is the matrix comprised of the rows of $W$ indexed by $\mathcal{A}$, and we denote the $i$-th row of $\hat{W}$ as $\hat{w}_i$, and the $i$-th entry of $\hat{b}$ as $\hat{b}_i$. Equation 6 follows since the elements of $W dz^{(d)}$ are i.i.d., and $\mathcal{A}^{(d)}$ selects all entries whose corresponding entries in $h^{(d)}$ have positive values. Thus, in expectation, pre-multiplying by the matrix $I_{\mathcal{A}^{(d)}}$ is equivalent to considering $\hat{W} dz^{(d)}$ instead of $I_{\mathcal{A}} W dz^{(d)}$ together with conditioning on the fact that every element in the vector $\hat{W} z^{(d)} + \hat{b}$ is positive.

This gives us

$$\mathbb{E}[\|I_{\mathcal{A}}Wdz^{(d)}\|\,] = \mathbb{E}\left[\underset{\hat{w}_1}{\mathbb{E}}\,\underset{\hat{w}_2}{\mathbb{E}}\cdots\underset{\hat{w}_{|\mathcal{A}|}}{\mathbb{E}}\left[\sqrt{\sum_{i=1}^{|\mathcal{A}|}(\hat{w}_i^\top dz^{(d)})^2}\ \middle|\ \hat{w}_i^\top z^{(d)} + \hat{b}_i > 0\ \forall i, |\mathcal{A}|\right]\right] \quad (7)$$

$$= \mathbb{E}\left[\underset{\hat{w}_1}{\mathbb{E}}\,\underset{\hat{w}_2}{\mathbb{E}}\cdots\underset{\hat{w}_{|\mathcal{A}|}}{\mathbb{E}}\left[\sqrt{\sum_{i=1}^{|\mathcal{A}|}|\hat{w}_i^\top dz^{(d)}|^2}\ \middle|\ \hat{w}_i^\top z^{(d)} + \hat{b}_i > 0\ \forall i, |\mathcal{A}|\right]\right] \quad (8)$$

$$\geq \mathbb{E}\left[\sqrt{\sum_{i=1}^{|\mathcal{A}|}\underset{\hat{w}_i}{\mathbb{E}}[|\hat{w}_i^\top dz^{(d)}|\,|\,\hat{w}_i^\top z^{(d)} + \hat{b}_i > 0]^2}\,\right], \quad (9)$$

where (7) follows from the analysis above and the independence of each $\hat{w}_i$, (8) is trivial, and (9) follows from iteratively applying Jensen's inequality, after noting that $f(x) = \sqrt{x^2 + C}$ is convex for $x, C \geq 0$.

Now let $J_i$ denote the (random) index set of the $\mathcal{P}$-distributed entries of $\hat{w}_i$, and let $w_{J_i}, dz_{J_i}^{(d)}, z_{J_i}^{(d)}$ denote the restrictions to the indices in $J_i$ of $\hat{w}_i$, $dz^{(d)}$ and $z^{(d)}$ respectively. Then $\hat{w}_i^\top z^{(d)} = w_{J_i}^\top z_{J_i}^{(d)}$, and $\hat{w}_i^\top dz^{(d)} = w_{J_i}^\top dz_{J_i}^{(d)}$, such that, after conditioning on $J_i$, we have that

$$\mathbb{E}[\|\hat{W}p\| \mid \hat{w}_i^\top z^{(d)} + \hat{b}_i > 0\ \forall i, |\mathcal{A}|] \geq \mathbb{E}\left[\underbrace{\sqrt{\sum_{i=1}^{|\mathcal{A}|}\underset{J_i}{\mathbb{E}}\Big[\overbrace{\underbrace{\underset{w_{J_i}}{\mathbb{E}}[|w_{J_i}^\top dz_{J_i}^{(d)}| \mid w_{J_i}^\top z_{J_i}^{(d)} + \hat{b}_i > 0, J_i]}_{(*)}}^{(**)}\Big]^2}}_{(***)}\right]. \quad (10)$$

**Stage 3:**

The third stage of the proof is to work our way from the inside out, lower bounding $(*)$ first, then $(**)$, and finally $(***)$.

Consider the expectation in $(*)$. Having conditioned on $J_i$, we can define $X = w_{J_i}^\top dz_{J_i}^{(d)}$ and $Y = w_{J_i}^\top z_{J_i}^{(d)} + \hat{b}_i$, such that lower bounding $(*)$ means lower bounding

$$\mathbb{E}[|X| \mid Y > 0]. \quad (11)$$

By assumption the joint distribution over $G = [w_{J_i,1}, \ldots, w_{J_i,k}, \hat{b}_i]^\top$ is even. The vector $H = [X, Y, w_{J_i,3}\ldots, w_{J_i,k}, \hat{b}_i]^\top$ is obtained by a linear transformation of $G$ (which is invertible since $\|z^{(d)}\|$ is not parallel to $\|dz^{(d)}\|$). Thus by Lemma 1 (continuous) or Lemma 2 (discrete) this joint distribution over $H$ is also even, and by Lemma 3 (continuous) or Lemma 4 (discrete), the joint distribution of $[X, Y]^\top$ is even too. We can therefore apply Lemma 5 (continuous) or Lemma 6 (discrete) and need only consider $\mathbb{E}[|X|]$, which is bounded as

$$\mathbb{E}[|X|] \geq M\|dz_{J_i}^{(d)}\|, \quad (12)$$

again by assumption.

Having bounded $(*)$, we average over $J_i$ to get $(**)$, for which we can apply Lemma 7 to get

$$\underset{J_i}{\mathbb{E}}[M\|dz_{J_i}^{(d)}\|] \geq \alpha M\|dz^{(d)}\|. \quad (13)$$

Finally, we can bound $(***)$ as follows

$$\mathbb{E}[\|I_{\mathcal{A}}W dz^{(d)}\|] \geq \underset{|\mathcal{A}|}{\mathbb{E}}\left[\sqrt{\sum_{i=1}^{|\mathcal{A}|} \alpha^2 M^2 \|dz^{(d)}\|^2}\right] \tag{14}$$

$$= \underset{|\mathcal{A}|}{\mathbb{E}}\left[\sqrt{|\mathcal{A}| \cdot \alpha^2 M^2 \|dz^{(d)}\|^2}\right] \tag{15}$$

$$\geq \underset{|\mathcal{A}|}{\mathbb{E}}\left[\frac{1}{\sqrt{k}\alpha M \|dz^{(d)}\|} \cdot |\mathcal{A}| \cdot \alpha^2 M^2 \|dz^{(d)}\|^2\right] \tag{16}$$

$$= \frac{\alpha M \|dz^{(d)}\|}{\sqrt{k}} \cdot \mathbb{E}[|\mathcal{A}|]. \tag{17}$$

where (14) is obtained by substituting the bound for $(**)$ into the inequality in (10), (15) follows since there is no dependence on $i$ in the summed terms, and (16) follows since for any $0 \leq \gamma \leq \max(\gamma)$, $\sqrt{\gamma} \geq \frac{1}{\sqrt{max(\gamma)}}\gamma$, and $|\mathcal{A}|$ is at most $k$.

The proof is concluded by calculating $\mathbb{E}[|\mathcal{A}|]$. Since $|\mathcal{A}|$ is the number of entries in the vector $h^{(d)}$ which are positive, and each entry in that vector is an independent, centred random variable, $|\mathcal{A}|$ has a binomial distribution with probability $1/2$, and therefore an expected value of $k/2$. Plugging this in yields the final recursive relation between $\|dz^{(d+1)}\|$ and $\|dz^{(d)}\|$,

$$\mathbb{E}[\|dz^{(d+1)}\|] \geq \frac{\alpha M \sqrt{k}}{2}\|dz^{(d)}\|.$$

Iterative application of this result starting at the first layer yields the final result.

$\square$

Let us illustrate the ease with which Corollaries 1, 2 and 3 are obtained. In the case of each distribution, we need to do two things. First, we must verify that the necessary assumption holds in the case of those distributions $\mathcal{P}$ and $\mathcal{Q}$: that the joint distribution over a vector of independent elements from both distributions is even. Second, we must derive a bound of the form $\mathbb{E}[|\boldsymbol{u}^\top \boldsymbol{w}|] \geq M\|\boldsymbol{u}\|$, where $w_i \sim \mathcal{P}$, and substitute $M$ into Theorem 2.

When $\mathcal{P}$ and $\mathcal{Q}$ are centred Gaussians, the joint distribution over elements from one or both distributions is a multivariate Gaussian, with an even joint probability density function. Moreover, for $U = \boldsymbol{u}^\top \boldsymbol{w}$, $\mathbb{E}[|U|]$ has a closed form solution,

$$\mathbb{E}[|U|] = \frac{\sqrt{2}\sigma_w}{\sqrt{\pi}}\|\boldsymbol{u}\|$$

When $\mathcal{P}$ and $\mathcal{Q}$ are centred uniform distributions, the joint distribution is uniform over the polygon bounded in each dimension by the symmetric bounds $[-C_w, C_w]$ or $[-C_b, C_b]$, and thus is even. Next, to bound $\mathbb{E}[|U|]$, we apply the Marcinkiewicz-Zygmund inequality with $p = 1$, using the optimal $A_1$ from Lemmas 8 and 9, to get that

$$\mathbb{E}[|U|] \geq \frac{C_w}{2\sqrt{2}}\|\boldsymbol{u}\|;$$

for details of this derivation, see Lemma 10.

Likewise, when $\mathcal{P}$ and $\mathcal{Q}$ are uniform distributions over discrete, symmetric, finite sets $\mathcal{W}$ and $\mathcal{B}$ respectively, we make a discrete analogue of the argument made in the continuous uniform case to confirm the necessary assumption holds. Bounding $\mathbb{E}[|U|]$ in this case also follows from a very similar argument to that made in the continuous case, detailed in full in Lemma 11, yielding

$$\mathbb{E}[|U|] \geq \frac{\sum_{w \in \mathcal{W}} |w|}{\sqrt{2}N_w}\|\boldsymbol{u}\|.$$

## 4 NUMERICAL SIMULATIONS

In this section we demonstrate, through numerical simulations, how the relationships between the the network's distributional and architectural properties observed in practice compare with those described in the lower bounds of Corollaries 1 - 3. To this end, we use as our trajectory a straight line between two (normalised) MNIST datapoints[1], discretized into 10000 pieces. For each combination of distribution and parameters, we pass the aforementioned line through 100 different deep neural networks of width $784$, and average the results. Specifically, we consider three different networks types, sparse-Gaussian, sparse-uniform, and sparse-discrete networks, from Definitions 2 - 4 respectively. For each distribution we consider different values of network fractional density $\alpha$ ranging from $0.1$ to $1$. In the sparse-Gaussian networks, non-zero weights are sampled from $\mathcal{N}(0, \sigma_w^2/k)$, and biases from $\mathcal{N}(0, 0.01^2)$. In the sparse-Uniform networks, non-zero weights are sampled from $\mathcal{U}(-C/\sqrt{k}, C/\sqrt{k})$, and biases from $\mathcal{U}(-0.01, 0.01)$. In the sparse-discrete networks, non-zero weights are uniformly sampled from $\mathcal{W} := (1/\sqrt{k}) \odot \{-C, -(C+1), \ldots, C-1, C\}$, and biases from $\mathcal{B} := \{-0.01, 0.01\}$. We do this for a variety of $\sigma_w$ and $C$ values. The results are shown in Figures 2 and 3.

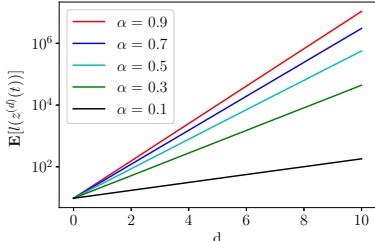

Figure 2: Expected length of a line connecting two MNIST data points as it passes through a sparse-Gaussian deep network, plotted at each layer $d$.

Figure 2 plots the average length of the trajectory at layer $d$ of a sparse-Gaussian network, with $\sigma_w = 6$ and for different choices of sparsity ranging from $0.1$ to $0.9$. We see exponential increase of expected length with depth even in sparse networks, with smaller slopes for smaller $\alpha$ (higher sparsity). In Figures 3a and 3b we plot the growth ratio of a small piece of the trajectory from one

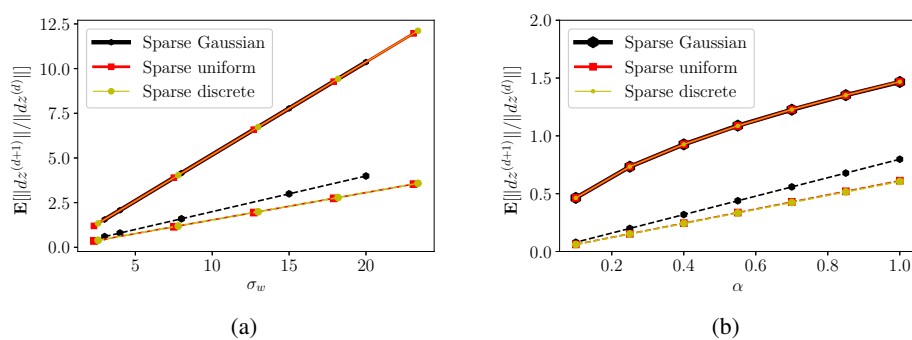

(a)                                                (b)

Figure 3: Expected growth factor, that is, the expected ratio of the length of any very small line segment in layer $d + 1$ to its length in layer $d$. Figure 3a shows the dependence on the variance of the weights' distribution, and Figure 3b shows the dependence on sparsity.

layer to the next, averaged over all pieces, at all layers, and across all 100 networks for a given distribution. This $\mathbb{E}[\|dz^{(d+1)}\|/\|dz^{(d)}\|]$ corresponds to the base of the exponential in our lower bound. The solid lines reflect the observed averages of this ratio, while the dashed lines reflect

---

[1]In this experiment we chose the $101^{st}$ and $1001^{st}$ points from the MNIST test set, but the choice of points does not qualitatively change the results.

the lower bound from Corollaries 1, 2, and 3. Figure 3a illustrates the dependence on the standard deviation of the respective distributions (before scaling by $1/\sqrt{k}$), with $\alpha$ fixed at $\alpha = 0.5$. We observe both that the lower bounds clearly hold, and that the dependence on $\sigma_w$ is linear in practice, exactly as we expect from our lower bounds. Figure 3b shows the dependence of this ratio on the sparsity parameter $\alpha$, where we have fixed $\sigma_w = 2$ for all distributions. Once again, the lower bounds hold, but in this case there is a slight curve in the observed values, not a strictly linear relationship. The reason for this is that the linear bound we provide is necessary in order to account for the more pathological cases of $dz$. This is discussed in more depth in Appendix B.

One striking observation in Figures 3a and 3b is that for a given $\sigma_w$, the observed $\mathbb{E}[\|dz^{(d+1)}\|/\|dz^{(d)}\|]$ matches perfectly across all three distributions, for different values of $\sigma_w$ and different $\alpha$. This remains true when we repeat the experiments with different datapoints, and with points chosen uniformly at random in a high-dimensional space, both when the trajectory considered is a straight line and when it is not (e.g. arcs in two or more dimensions.) See Appendix C.1 for these figures. Another implication of these experiments is that they give some guidance for how to trade off weight scale against sparsity depending on the desired network properties. For example, Figure 3b considers the initialisation scheme with $\sigma_w = 2/\sqrt{k}$. We see that the empirically observed growth factor from one layer to the next is approximately 1.5 when the matrices are dense ($\alpha = 1$), while the growth factor is 1 with $\alpha \approx 0.5$, and less than one as $\alpha$ decreases further.

In Appendix C.2 we present some preliminary numerical experiments looking at the extent to which the results proved and verified for random networks apply to trained networks as well. Our results indicate that indeed some of the results appear to carry over: trained networks appear to retain the exponential trajectory growth with depth, and that the trajectory growth factor retains a roughly linear dependence on $\sigma_w$. However, this expected growth factor is not the same size in random nets as it is in trained nets with the same $\sigma_w$, nor is the expected growth factor trajectory independent in trained networks, as it is for random networks.

## 5 CONCLUSION

Our proof strategy and results generalise and extend previous work by Raghu et al. (2017) to develop theoretical guarantees lower bounding expected trajectory growth through deep neural networks for a broader class of network weight distributions and the setting of sparse networks. We illustrate this approach with Gaussian, uniform, and discrete valued random weight matrices with any sparsity level.

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

## A  SUPPORTING LEMMAS

**Lemma 1.** *Let $f_X(\mathbf{x})$ be an even joint probability density function over random vector $X \in \mathbb{R}^k$. Let $A \in \mathbb{R}^{k \times k}$ be an invertable linear transformation such that $Y = AX$. Then the joint density $f_Y(\mathbf{y})$ is also even.*

*Proof.* Wlog we assume $f_X$ is defined on $\mathbb{R}^k$. To calculate the density over $Y \in \mathbb{R}^k$ we make a change of variables such that

$$f_Y(\mathbf{y}) = f_X(A^{-1}\mathbf{y})|A^{-1}|. \tag{18}$$

Since $A$ is one-to-one, we have that $f_X(\boldsymbol{x}) = f_X(A^{-1}\boldsymbol{y})$ for some $\boldsymbol{y}$, and $f_X$ is even, so $f_X(A^{-1}\boldsymbol{y}) = f_X(-(A^{-1}\boldsymbol{y})) = f_X(A^{-1}(-\boldsymbol{y}))$ for all $\boldsymbol{y}$. Putting this together completes the proof,

$$f_Y(\mathbf{y}) = f_X(A^{-1}\mathbf{y})|A^{-1}| = f_X(A^{-1}(-\mathbf{y}))|A^{-1}| = f_Y(-\mathbf{y}) \tag{19}$$

$\square$

**Lemma 2.** *Let $f_X(\mathbf{x})$ be an even joint probability mass function over random vector $X \in \mathbb{R}^k$. Let $A \in \mathbb{R}^{k \times k}$ be an invertable linear transformation such that $Y = AX$. Then the joint mass function $f_Y(\mathbf{y})$ is also even.*

*Proof.* $f_X$ is defined on some discrete, finite, symmetric set $\mathcal{X}$. To calculate the density over $Y \in \mathcal{Y} := \{Ap : p \in \mathcal{X}\}$ we make a change of variables such that

$$f_Y(\mathbf{y}) = \sum_{\boldsymbol{x} \in \{A\boldsymbol{x}=\boldsymbol{y}\}} f_X(\mathbf{x}). \tag{20}$$

Since $A$ is one-to-one, we have that $f_X(\boldsymbol{x}) = f_X(A^{-1}\boldsymbol{y})$ for some $\boldsymbol{y}$, and $f_X$ is even, so $f_X(A^{-1}\boldsymbol{y}) = f_X(-(A^{-1}\boldsymbol{y})) = f_X(A^{-1}(-\boldsymbol{y}))$ for all $\boldsymbol{y}$. Putting this together completes the proof,

$$f_Y(\mathbf{y}) = \sum_{\boldsymbol{x} \in \{A\boldsymbol{x}=\boldsymbol{y}\}} f_X(A^{-1}\boldsymbol{y}) = \sum_{\boldsymbol{x} \in \{A\boldsymbol{x}=\boldsymbol{y}\}} f_X(A^{-1}(-\boldsymbol{y})) = f_Y(-\mathbf{y}) \tag{21}$$

$\square$

**Lemma 3.** *Let $f_{X_1,\dots,X_k}(x_1,\dots,x_k)$ be an even probability density function. Then $f_{X_1,\dots,X_{k-1}}(x_1,\dots,x_{k-1}) = \int_{-\infty}^{\infty} f_{X_1,\dots,X_k}(x_1,\dots,x_k)dx_k$ is also even.*

*Proof.*

$$
\begin{aligned}
f_{X_1,\dots,X_{k-1}}(x_1,\dots,x_{k-1}) &= \int_{-\infty}^{\infty} f_{X_1,\dots,X_k}(x_1,\dots,x_k)dx_k \\
&= \int_{-\infty}^{\infty} f_{X_1,\dots,X_k}(-x_1,\dots,-x_k)dx_k \\
&= \int_{-\infty}^{\infty} f_{X_1,\dots,X_k}(-x_1,\dots,-x_{k-1},x_k)dx_k \\
&= f_{X_1,\dots,X_{k-1}}(-x_1,\dots,-x_{k-1})
\end{aligned}
$$

The first and last equalities follow from the definition of marginalisation of random variables. The second equality follows from the assumption that $f_{X_1,\dots,X_k}$ is even, and the third equality follows from the change of variables: $-x_k \longrightarrow x_k$.  $\square$

**Lemma 4.** *Let $X_1,\dots,X_k$ be discrete random variables with symmetric support sets $\mathcal{X}_1,\dots,\mathcal{X}_k$ respectively, i.e. $x_i \in \mathcal{X}_j \iff -x_i \in \mathcal{X}_j$. Let $P(X_1 = x_1,\dots,X_k = x_k)$ be an even probability mass function such that $P(X_1 = x_1,\dots,X_k = x_k) = P(X_1 = -x_1,\dots,X_k = -x_k)$.*

*Then $P(X_1 = x_1,\dots,X_{k-1} = x_{k-1})$ is also even.*

*Proof.*

$$P(X_1 = x_1, \ldots, X_{k-1} = x_{k-1}) = \sum_{x_k \in \mathcal{X}_k} P(X_1 = x_1, \ldots, X_k = x_k) \tag{22}$$

$$= \sum_{x_k \in \mathcal{X}_k} P(X_1 = -x_1, \ldots, X_k = -x_k) \tag{23}$$

$$= \sum_{-x_k \in \mathcal{X}_k} P(X_1 = -x_1, \ldots, X_k = x_k) \tag{24}$$

$$= \sum_{x_k \in \mathcal{X}_k} P(X_1 = -x_1, \ldots, X_k = x_k) \tag{25}$$

$$= P(X_1 = -x_1, \ldots, X_{k-1} = -x_{k-1}) \tag{26}$$

Lines 22 and 26 follow from the definition of marginal distributions, (23) follows by assumption, (24) follows fro a change of variables, and (25) follows since summing over $-x_k$ is equivalent to summing over $x_k$.

$\square$

**Lemma 5.** *Let $X$ and $Y$ be random variables with an even joint probability density function $f_{XY}(x, y)$. Then*

$$\mathbb{E}[|X| \mid Y > 0] = \mathbb{E}[|X|]$$

*Proof.* Letting $|X| = Z$, we can make a straightforward change of variables to calculate the joint distribution $f_{ZY}(z, y)$, which works out to be

$$f_{ZY}(z, y) = f_{XY}(z, y) + f_{XY}(-z, y)$$

for $z \geq 0$ and $y \in \mathbb{R}$. Then we have that

$$\mathbb{E}[Z|Y > 0] = \int_0^\infty z \cdot f_{Z|Y>0}(z|y > 0)dz$$

$$= \int_0^\infty z \cdot \frac{f_{Z,Y>0}(z, y > 0)}{\int_0^\infty f_Y(y)dy} dz$$

$$= 2 \int_0^\infty z \cdot f_{Z,Y>0}(z, y > 0)dz$$

$$= 2 \int_0^\infty z \int_0^\infty f_{ZY}(z, y)dydz$$

$$= 2 \int_0^\infty z \int_0^\infty (f_{XY}(z, y) + f_{XY}(-z, y))dydz.$$

One the other hand, we have that

$$\mathbb{E}[Z] = \int_0^\infty z \cdot f_Z(z)dz$$

$$= \int_0^\infty z \cdot (f_X(z) + f_X(-z))dz$$

$$= 2 \int_0^\infty z \cdot f_X(z)dz$$

$$= 2 \int_0^\infty z \cdot \int_{-\infty}^\infty f_{XY}(z, y)dydz$$

$$= 2 \int_0^\infty z \cdot \left( \int_{-\infty}^0 f_{XY}(z, y)dy + \int_0^\infty f_{XY}(z, y)dy \right) dz$$

Comparing the expressions for $\mathbb{E}[Z|Y > 0]$ and $\mathbb{E}[Z]$, we can see that they are equal if

$$\int_{-\infty}^0 f_{XY}(z, y) dy = \int_0^\infty f_{XY}(-z, y) dy.$$

A change of variables on the left hand side from $y$ to $-y$ yields

$$\int_{-\infty}^0 f_{XY}(z, y) dy = \int_0^\infty f_{XY}(z, -y) dy.$$

and by assumption, we know that $f_{XY}(z, -y) = f_{XY}(-z, y)$ since $f_{XY}$ is even, which completes the proof.

$\square$

Lemma 5 implicitly makes use of the fact that $P(Y = 0) = 0$, which follows from $w_{J_i}$ and $\hat{b}_i$ being continuous random variables, and $Y = w_{J_i}^\top z_{J_i} + \hat{b}_i$, with $z_{J_i}$ being fixed independent of $w_{J_i}$. We similarly make use of the fact that $P(Y = 0) = 0$ in the application of Lemma 6, though that this is true is less immediately apparent in the discrete case. For clarity, let us define $v := [w_{J_i}, \hat{b}_i]$, the concatenation of $w_{J_i}$ and $\hat{b}_i$, and $\hat{z} := [z_{J_i}, 1]$, the concatenation of $z_{J_i}$ and 1, such that $Y = v^\top \hat{z}$. Associated with the discrete distribution over $v$ there are $N_w^{|J_i|} N_b$ possible discrete random vectors in $\mathbb{R}^{|J_i|+1}$. The set of vectors $\hat{z} \in \mathbb{R}^{|J_i|+1}$ orthogonal to such a discrete set is measure zero, and as such for $\hat{z}$ fixed independent of the choice of the discrete measure $v$ we have $P(v^\top \hat{z} = 0) = 0$. If however $\hat{z}$ were selected with knowledge of the discrete distribution $v$ then one of two cases will occur; either $v^\top \hat{z} \neq 0$, or $\hat{z}$ is selected to be from the measure zero set of vectors orthogonal to any of the $N_w^{|J_i|} N_b$ vectors generated by $v$. In the latter case, the assumptions in Lemma 6 of $\mathcal{Y}$ excluding 0 would not be satisfied. In such an adversarial case there would be a discrepancy between $\mathbb{E}[|X| \mid Y > 0]$ and $\mathbb{E}[|X|]$ which would shrink as the proportion of the $N_w^{|J_i|} N_b$ vectors generated by $v$ to which that particular $\hat{z}$ is orthogonal.

**Lemma 6.** *Let $X$ and $Y$ be discrete random variables with finite, symmetric support sets $\mathcal{X}$ and $\mathcal{Y}$ respectively, where $0 \notin \mathcal{Y}$, and an even joint probability mass function $f_{XY}(x, y)$ such that $P(X = x, Y = y) = P(X = -x, Y = -y)$. Then*

$$\mathbb{E}[|X| \mid Y > 0] = \mathbb{E}[|X|]$$

*Proof.* Letting $|X| = Z$, we can make a change of variables to obtain the joint mass function $f_{ZY}(z, y)$, which works out to be

$$f_{ZY}(z, y) = \begin{cases} f_{XY}(z, y) + f_{XY}(-z, y) & \text{for } (z, y) \text{ where } z \in \mathcal{X}^+ \text{ and } y \in \mathcal{Y} \\ f_{XY}(z, y) & \text{for } (z, y) \text{ where } z = 0 \text{ and } \in \mathcal{Y} \end{cases}$$

where $\mathcal{X}^+$ is the set of all positive elements of $\mathcal{X}$.

Next, we have that

$$\begin{aligned}
\mathbb{E}[Z|Y > 0] &= \sum_{z \in \mathcal{X}^+} z P(Z = z | Y > 0) \\
&= \sum_{z \in \mathcal{X}^+} z \frac{P(Z = z \cap Y > 0)}{P(Y > 0)} & (27) \\
&= 2 \sum_{z \in \mathcal{X}^+} z P(Z = z \cap Y > 0) & (28) \\
&= 2 \sum_{z \in \mathcal{X}^+} \sum_{y \in \mathcal{Y}^+} z P(Z = z \cap Y = y) \\
&= 2 \sum_{z \in \mathcal{X}^+} \sum_{y \in \mathcal{Y}^+} z \left( f_{XY}(z, y) + f_{XY}(-z, y) \right) & (29)
\end{aligned}$$

On the other hand, we have

$$\mathbb{E}[Z] = \sum_{z \in \mathcal{X}^+} z P(Z = z) \tag{30}$$

$$= \sum_{z \in \mathcal{X}^+} z \left(f_X(z) + f_X(-z)\right) \tag{31}$$

$$= 2 \sum_{z \in \mathcal{X}^+} z f_X(z) \tag{32}$$

$$= 2 \sum_{z \in \mathcal{X}^+} \sum_{y \in \mathcal{Y}} z f_{XY}(z, y) \tag{33}$$

$$= 2 \sum_{z \in \mathcal{X}^+} \left( \sum_{y \in \mathcal{Y}^+} z f_{XY}(z, y) + \sum_{y \in \mathcal{Y}^-} z f_{XY}(z, y) \right) \tag{34}$$

Next, we not that

$$\sum_{y \in \mathcal{Y}^-} z f_{XY}(z, y) = \sum_{y \in \mathcal{Y}^+} z f_{XY}(z, -y)$$

$$= \sum_{y \in \mathcal{Y}^+} z f_{XY}(-z, y)$$

Thus the expressions in 29 and 34 are equal, which completes the proof.

$\square$

**Lemma 7** (Expected norm of a random sub-vector). *Let $\boldsymbol{u} \in \mathbb{R}^k$ be a fixed vector and let $J \subseteq \{1, 2, \ldots, k\}$ be a random index set, where the probability of any index from 1 to $k$ appearing in any given sample is independent and equal to $\alpha$. Then, defining $\boldsymbol{u}_J$ to be the vector comprised only of the elements of $\boldsymbol{u}$ indexed by $J$, we can lower bound the expectation of the norm of this subvector by*

$$\mathbb{E}_J[\|\boldsymbol{u}_J\|] \geq \alpha \|\boldsymbol{u}\| \tag{35}$$

*Proof.* First, we bound the expectation of the norm in terms of the expectation of the squared norm as follows:

$$\mathbb{E}[\|\boldsymbol{u}_J\|] = \mathbb{E}\left[\sqrt{\sum_{j \in J} u_{J,j}^2}\right] \tag{36}$$

$$\geq \frac{1}{\|\boldsymbol{u}\|} \mathbb{E}\left[\sum_{j \in J} u_{J,j}^2\right] \tag{37}$$

This follows because for any $0 \leq \gamma \leq \max(\gamma)$, $\sqrt{\gamma} \geq \frac{1}{\sqrt{max(\gamma)}} \gamma$.

Next we note that $\sum_{j \in J} u_{J,j}^2$ is exactly equivalent to $\sum_{i=1}^{k} u_i^2 B_i$, a weighted sum of $k$ iid Bernoulli random variables $B_i$ with $p = \alpha$, and so

$$\mathbb{E}\left[\sum_{j \in J} u_{J,j}^2\right] = \sum_{i=1}^{k} u_i^2 \cdot \mathbb{E}[B] \tag{38}$$

$$= \|\boldsymbol{u}\|^2 \cdot \alpha. \tag{39}$$

Substituting this into inequality 37 completes the proof,

$$\mathbb{E}[\|\boldsymbol{u}_J\|] \geq \alpha \|\boldsymbol{u}\|$$

$\square$

Lemmas 8 and 9 are taken from Ferger (2014), and are restated here for completeness.

**Lemma 8** (Marcinkiewicz-Zygmund Inequality (Ferger (2014))). *Let $X_1, \ldots, X_n$ be $n \in \mathbb{N}$ independent and centered real random variables defined on some probability space $(\Omega, A, P)$ with $\mathbb{E}[|Xi|^p] < \infty$ for every $i \in \{1, \ldots, n\}$ and for some $p > 0$. Then for every $p \geq 1$ there exist positive constants $A_p$ and $B_p$ depending only on $p$ such that*

$$A_p \mathbb{E}\left[\left(\sum_{i=1}^n X_i^2\right)^{p/2}\right] \leq \mathbb{E}\left[\left|\sum_{i=1}^n X_i\right|^p\right] \leq B_p \mathbb{E}\left[\left(\sum_{i=1}^n X_i^2\right)^{p/2}\right] \tag{40}$$

**Lemma 9** (Optimal constants for Marcinkiewicz-Zygmund Inequality (Ferger (2014))). *Let $\Gamma$ denote the Gamma function and let $p_0$ be the solution of the equation $\Gamma(\frac{p+1}{2}) = \sqrt{\pi}/2$ in the interval $(1, 2)$, i.e. $p_0 \approx 1.84742$. Then for every $p > 0$ it holds:*

$$A_{p,opt} = \begin{cases} 2^{p/2-1}, & 0 < p \leq p_0 \\ 2^{p/2} \cdot \frac{\Gamma\left(\frac{p+1}{2}\right)}{\sqrt{\pi}}, & p_0 < p < 2 \\ 1 & 2 \leq p < \infty \end{cases} \tag{41}$$

*and*

$$B_{p,opt} = \begin{cases} 1 & 0 < p \leq 2 \\ 2^{p/2} \cdot \frac{\Gamma\left(\frac{p+1}{2}\right)}{\sqrt{\pi}}, & 2 < p < \infty \end{cases} \tag{42}$$

**Lemma 10.** *Let $X = \sum_i \alpha_i w_i$, where $w_i \sim \mathcal{U}(-C, C)$ Then*

$$\mathbb{E}[|X|] \geq \frac{C}{2\sqrt{2}}\|\alpha\|$$

*Proof.* Defining $X_i = \alpha_i w_i$, we can then apply the Marcinkiewicz-Zygmund inequality with $p = 1$, using the optimal $A_1$ from Lemma 9 to get that

$$\mathbb{E}[|X|] = \mathbb{E}\left[\left|\sum_{i=1}^k X_i\right|\right] \geq \frac{1}{\sqrt{2}}\mathbb{E}\left[\sqrt{\sum_{i=1}^k X_i^2}\right]$$

Next we use the same tricks as early in the proof of the Gaussian case:

$$\frac{1}{\sqrt{2}}\mathbb{E}\left[\sqrt{\sum_{i=1}^k X_i^2}\right] = \frac{1}{\sqrt{2}}\mathbb{E}\left[\sqrt{\sum_{i=1}^k |X_i|^2}\right]$$

$$\geq \frac{1}{\sqrt{2}}\sqrt{\sum_{i=1}^k \mathbb{E}[|X_i|]^2},$$

where the first equality is trivial and the second follows from a repeated application of Jensen's inequality.

To calculate $\mathbb{E}[|X_i|]$ we note that $X_i = \alpha_i w_i$ is uniformly distributed $X_i \sim U(-|\alpha_i|C, |\alpha_i|C)$, and thus

$$\mathbb{E}[|X_i|] = \frac{C|\alpha_i|}{2}$$

and so

$$\mathbb{E}[|X|] \geq \frac{1}{\sqrt{2}}\sqrt{\sum_{i=1}^k \mathbb{E}[|X_i|]^2}$$

$$= \frac{1}{\sqrt{2}}\sqrt{\frac{C^2}{4}\sum_{i=1}^k |\alpha_i|^2}$$

$$= \frac{C}{2\sqrt{2}}\|\alpha\|$$

$\square$

**Lemma 11.** *Let $X = \sum_i \alpha_i w_i$, where $w_i$ are uniformly sampled from some discrete symmetric sample space $\mathcal{W}$. Then*

$$\mathbb{E}[|X|] \geq \frac{\sum_{w \in \mathcal{W}} |w|}{\sqrt{2} N_w} \|\alpha\|$$

*Proof.* Defining $X_i = \alpha_i w_i$, we follow exactly the same steps as in the first part of the proof of Lemma 10, to get that

$$\mathbb{E}[|X|] \geq \frac{1}{\sqrt{2}} \sqrt{\sum_{i=1}^{k} \mathbb{E}[|X_i|]^2}.$$

To calculate $\mathbb{E}[|X_i|]$ we note that $X_i = \alpha_i w_i$ is uniformly sampled from $\alpha_i \mathcal{W}$ and thus

$$\mathbb{E}[|X_i|] = \frac{|\alpha_i| \sum_{w \in \mathcal{W}} |w|}{N_w}$$

and so

$$\begin{aligned}
\mathbb{E}[|X|] &\geq \frac{1}{\sqrt{2}} \sqrt{\sum_{i=1}^{k} \mathbb{E}[|X_i|]^2} \\
&= \frac{1}{\sqrt{2}} \sqrt{\frac{(\sum_{w \in \mathcal{W}} |w|)^2}{N_w^2} \sum_{i=1}^{k} |\alpha_i|^2} \\
&= \frac{\sum_{w \in \mathcal{W}} |w|}{\sqrt{2} N_w} \|\alpha\|
\end{aligned}$$

$\square$

**Lemma 12.** *Let $\mathcal{W}, \mathcal{X} \subset \mathbb{R}^k$ be discrete sets with finite cardinality, and $g : \mathcal{W} \longrightarrow \mathcal{X}$ be a one-to-one transformation. Then if $P(W = \mathbf{w}) = P(W_1 = w_1, \ldots, W_k = w_k) = C$ for all $\mathbf{w} \in \mathcal{W}$, where $C$ is constant, then $P(X = \mathbf{x}) = C$ for all $\mathbf{x} \in \mathcal{X}$*

*Proof.*

$$P(X = \mathbf{x}) = \sum_{\mathbf{w} \in \{g(\mathbf{w}) = \mathbf{x}\}} P(W = \mathbf{w}) \tag{43}$$

$$= C \tag{44}$$

Equation 43 is a change of variables, and (44) follows from the fact the there is only ever one term in the sum, since $g$ is one-to-one. $\square$

## B  Non-linear dependence on $\alpha$ in the typical case

One interesting observation which merits further detail is that the observed dependence of the growth factor on $\alpha$ in practice, shown in Figure 3b, is not exactly linear, but rather the shape of that dependence looks closer to $\sqrt{\alpha}$. The likely source of this qualitative discrepancy is the use of Lemma 7, to lower bound

$$\mathbb{E}_{J_i}[\|dz_{J_i}\|] \geq \alpha \|dz\|, \tag{45}$$

used in (13) in Stage 3 of the proof of Theorem2. It is straightforward to derive an *upper* bound for this same quantity, as

$$\mathbb{E}_{J_i}[\|dz_{J_i}\|] \leq \sqrt{\alpha} \|dz\|, \tag{46}$$

first using Jensen's inequality to get that $\mathbb{E}_{J_i}[\sqrt{\|dz_{J_i}\|^2}] \leq \sqrt{\mathbb{E}[\|dz_{J_i}\|^2]}$, and then using the strategy from the proof of Lemma 7 to get $\mathbb{E}[\|dz_{J_i}\|^2] = \alpha \|dz\|^2$.

To explore this discrepancy between the observed growth ratio and the lower and upper bounds from (45) and (46), we consider different fixed vectors $dz \in \mathbb{R}^k$, and average over subvectors $dz_{J_i}$. Specifically, we calculated the expected value of a subvector $dz_{J_i}$ containing only the entries of $dz$ indexed by $J_i$, where $J_i \subseteq \{1, 2, \ldots, k\}$ is a random index set, where the probability of any index from 1 to $k$ appearing in any given sample is independent and equal to $\alpha$. Figure 4a shows the results when $dz$ a realisation of the uniform distribution over the unit sphere, with different dimensions $k$.

For even moderately large k, and vectors $dz$ where most entries are roughly this same magnitude, this upper bound is very tight, such that the expected norm of the subvector generally behaves like $\sqrt{\alpha}\|dz\|$, not $\alpha\|dz\|$. However, it is also possible to construct an example where the lower bound is

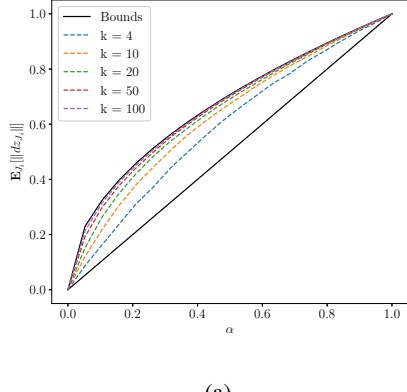
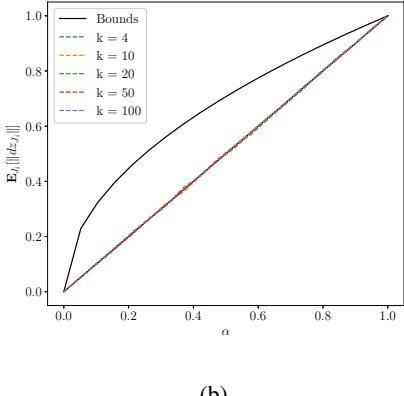

(a)                                    (b)

Figure 4: The dependence on $\alpha$ and $k$ of expected value of a subvector $dz_{J_i}$. In Figure 4a, $dz$ is a realisation of the uniform distribution over the unit sphere. In Figure 4b, $dz$ has the first entry equal to 1, and the rest zeros.

tight, by letting $dz$ have only a single non-zero entry, which case $\mathbb{E}[\|\boldsymbol{u}_J\|] = \alpha\|\boldsymbol{u}\|$ (see Figure 4b). While the former case, with entries of $dz$ mostly of the same order, is typical, especially past the first few layers of the network, the bound cannot be improved without further assumptions on $\|dz\|$. Further work on quantifying the probabilistic concentration of $\mathbb{E}[\|\boldsymbol{u}_J\|]$ close to $\sqrt{\alpha}\|\boldsymbol{u}\|$ would be an interesting extension of this research.

## C  ADDITIONAL NUMERICAL EXPERIMENTS

### C.1  OTHER DATAPOINTS AND TRAJECTORIES WITH RANDOM NETWORKS

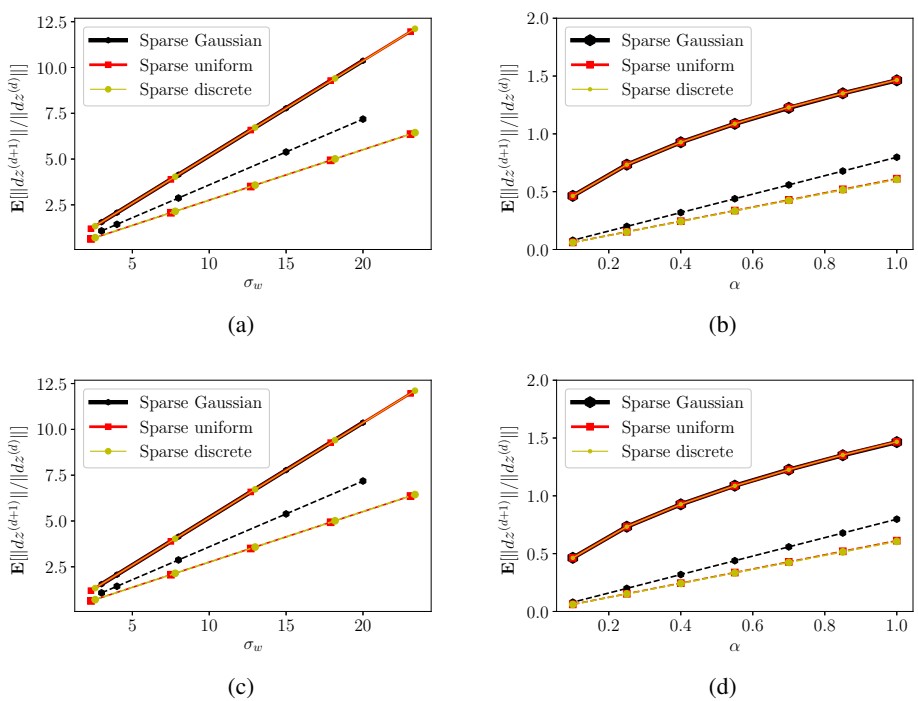

(a)

(b)

(c)

(d)

Figure 5: Expected growth factor for trajectories joining randomly chosen (normalised) points in $\mathbb{R}^{500}$. Figure 5a and Figure 5c show the dependence on the standard deviation of the weights' distribution for a straight and curved trajectory respectively, and Figure 5b and Figure 5d show the dependence on sparsity with a straight and curved trajectory respectively. In this experiment we have chosen as the curved trajectory a straight line which has been modified to be a semi-circular arc in 100 randomly chosen hyperplanes.

### C.2  TRAINED NETWORKS

One might consider the extent to which the results proved and numerically verified for random networks carry over to the context of trained networks.

Here we present a preliminary investigation into trajectory growth through trained networks. For 5 different values of initial $\sigma_w$, we train 20 different randomly initialised dense Gaussian deep ReLU networks (10 layers, hidden layers of width 32) on MNIST to test accuracy of greater than $95\%$. After training we calculate the new, post-training $\sigma_w$ of each net. We then sample pairs of data points of the same class, and of different classes, and connect them with straight lines. These are the trajectories we pass through the trained networks (each is pass through the 20 trained networks for each $\sigma_w$, and the growth factor averaged over all these networks). Each trajectory was then also passed through 20 random Gaussian networks of the same architecture in order to compare the outcomes.

The results are shown in Figure 6. Two key observations from Figure 6, consistent with Corollaries 1 - 3, are that: there remains a roughly linear dependence of $\sigma_w$ in the base of the growth factor and 2) that the expected trajectory growth remains exponential with depth.

Figure 6a illustrates this for paths between data points within and between classes, which is one of the potential factors that would impact the behaviour of a trained, rather than random, network; we observe that the average growth factor for trajectories between classes is larger than for trajectories

joining points within the same class. However, trajectory growth can remain exponential in depth in both cases.

Figure 6b shows individual results for individual lines passed through the 20 trained networks per initialization value $\sigma_w$. In addition to the average behaviour shown in Figure 6a, we observe that each path shows the same two aforementioned key observations of path length having dependence proportional to $\sigma_w$. The black curves in Figure 6b are the result of the same procedure, but with random networks pre-training. Individual data points in Figure 6b come from averaging the post-training $\sigma_w$ and trajectory growth factor over twenty trained networks, each curve in Figure 6b thus corresponds to one trajectory, for varying values of $\sigma_w$. The variance, over the 20 trained networks, of both computed values (post-training $\sigma_w$ and growth factor), is displayed in the figures, but is typically smaller than the dot used by the plotting software. Consequently visual gaps between data points indicate a statistically significant difference between the expected growth factors for different trajectories. The consistency across different trained networks for each given trajectory is remarkable given the different random initializations of the networks. Moreover, however, the gaps between the plots for different trajectories illustrates a key different between trained and untrained networks: in trained networks, the growth factor is trajectory-dependent, while (as shown in Figure 5), this is not the case for random networks.

A further numerical exploration of these preliminary results would be necessary to confirm consistency across the many hyperparameters involved in such experiments; for example, which sample pairs of points are used, what trajectory shape, or what regularisation is used during training. Similarly, potential experiments on trained sparse networks may be influenced by the choice of method for imposing sparsity on the trained network (e.g variational dropout (Molchanov et al. (2017)), $l_0$ regularization (Louizos et al. (2017)), or magnitude pruning (Zhu & Gupta (2017))). Such an expansive numerical investigation is beyond the scope of the focus of this paper, but may serve as an interesting alternative direction of inquiry.

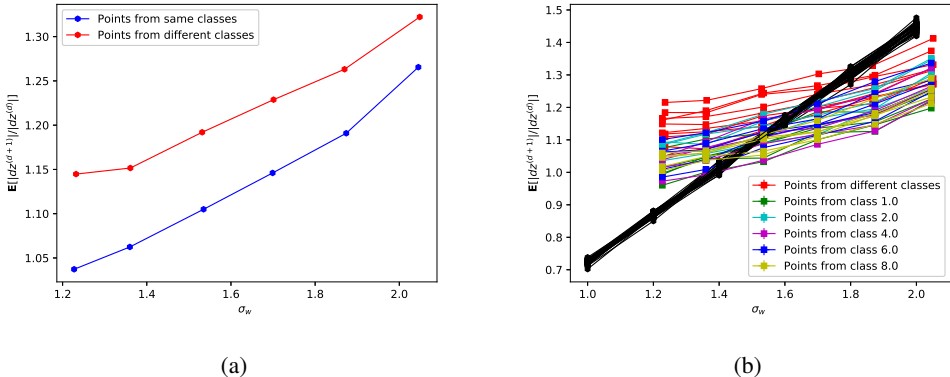

(a)  (b)

Figure 6: Expected growth factor for trajectories passed through trained feedforward ReLU networks trained on MNIST.

