# OpenReview forum: "Trajectory growth through random deep ReLU networks"
_ICLR.cc/2020/Conference — Reject_

### Official Review · AnonReviewer2 · 2019-10-15
**Official Blind Review #2**

**Rating:** 3

**Review:**

In this paper, how the length of the trajectory in the input space is amplified by a ReLU neural network is analyzed. Specifically, the paper studied the case when the weights and biases are sparse random matrices. Some theoretical lower bounds are derived and are also empirically checked.

Though the results are interesting, I am slightly lean to the rejection side. The main reason is that the motivation is not strong enough and it makes the entire work somehow incremental.

As described in the paper, the analysis of the trajectory length of NN has been initiated by Raghu et al. (2017). Although Raghu et al. considered the case of densely connected NNs, this work extended the notion to the sparsely connected NNs. I feel the extension is reasonable but it is not a big "jump" from the original work, and the originality is less significant. A good point of the derived results is that they are simple and easy to understand the dependency of the variance of the weights and the sparsity level. However, I am not so excited about these results because I cannot find a practical value from them. For example, (the current form of) Corollary 1 does not tell us how to control the sparsity level \alpha to maintain some accuracy-sparsity tradeoff.

From the technical side, it is nice the proof is written in line by line. However, it looks everything is done in a normal manner and nothing special happens. Also, the condition of Theorem 2 is unclear. What does the "restriction" of \hat{w}_i mean? Does the condition E[] >= M ||u|| hold for any u and M?

Finally, I have a simple question about the relationship between the scale of the output and the trajectory length. Let W be a weight and x be an input. When the smallest singular value of W is m, x is amplified at least by m, i.e., ||Wx|| >= m||x|| for any x. This means that (a part of) the trajectory growth is explained by the smallest singular value of the sparse random matrix W. Can you clarify the difference?

**Experience Assessment:**

I do not know much about this area.

**Review Assessment: Checking Correctness Of Derivations And Theory:**

I assessed the sensibility of the derivations and theory.

**Review Assessment: Checking Correctness Of Experiments:**

I assessed the sensibility of the experiments.

**Review Assessment: Thoroughness In Paper Reading:**

I read the paper at least twice and used my best judgement in assessing the paper.

---

> ### Author Response · Authors · 2019-11-08
> **Response to Review #2, Part 1**
>
> Thank you for your detailed reading of our work, and for your positive comments regarding the clarity of our results.
>
> Let us respond to the specific concerns listed about our paper.
>
> •	“The motivation is not strong enough and it makes the entire work somehow incremental.”; “I feel the extension is reasonable but it is not a big "jump" from the original work, and the originality is less significant.”
>
> Allow us to motivate the significance and innovation of our work further. The crucial point is that even after you read and understand the prior work by Raghu et al., there is no way to use their approach to achieve a more general result like ours. As such our work is not simply an extension of their work or their proof, but rather a proof of a distinct, more general result, albeit inspired by the idea of considering trajectory growth (for which they deserve full credit). To be precise, the innovation of our proof is the strategy for explicitly accounting for the statistical dependence of an infinitesimal piece of the trajectory $dz$ on $z$ in such a way which allows us to prove a more general and more precise result. Conversely, the key fundamentals of the proof by Raghu et al. *limit* extension to more general cases: In particular, their proof is based on splitting the components of $dz$ into parts which are perpendicular and parallel to $z$ (which we do not do). One consequence of this strategy is that it requires that, after splitting the weight matrix into parallel and perpendicular parts $W_\top$ and $W_\|$, that these matrices are independent random matrices (their lemma 2). Everything from this point on in their proof relies on this fact – and *this is only true for Gaussian random matrices*. Thus, their proof strategy has no clear extension to sparse matrices and other distributions. Ours is not an extension *of their proof*, but rather is a completely different method to derive more general and more accurate bounds. A consequence of this is that in the specific standard Gaussian case, our work provides an alternative and more straightforward proof of a similar result to theirs. We apologise if in attempting to ‘give credit where credit is due’, we did not make clear enough in our writing the extent to which Raghu et al.’s work did not in any way constitute or provide a blueprint for the key innovation of this work, but rather was the inspiration and the first such bounds in the case of dense Gaussian weight matrices.
>
> Finally, it is only by considering examples from this more general class of distributions that we empirically observe the common and exclusive dependence of length growth on standard deviation and sparsity level, independent of distribution choice (Figures 3a and 3b).
>
> •	“However, I am not so excited about these results because I cannot find a practical value from them. For example, (the current form of) Corollary 1 does not tell us how to control the sparsity level $\alpha$ to maintain some accuracy-sparsity tradeoff.“
>
> The bounds show how one can trade-off between weight variance and sparsity so as to control lower bounds on the expected length growth.  When the base in lower bound is greater than one the network will necessarily generate structures with exponential growth, even if that is expressly not desired.  The bounds give formulae by which a user can adjust these parameters to, potentially, avoid this.  For instance, in the case of sparse-uniform initialisation, with uniform weights sampled from $U(-10/\sqrt{k}, 10/\sqrt{k})$, then if one sets the sparsity parameter $\alpha < 0.56$, the lower bound will  converge to zero and the network need not (but might as this is a lower bound) have an exponential growth in  trajectory length. To the best of our knowledge no such information was previously available to practitioners. Moreover, our numerical experiments provide further practical insight for practitioners. For example, Figure 3b considers the initialisation scheme with $\sigma_w = 2/\sqrt{k}$. We see that the empirically observed growth factor from one layer to the next is approximately 1.5 when the matrices are dense ($\alpha=1$), while the growth factor is 1 with $\alpha \approx 0.5$, and less than one as $\alpha$ decreases further. This gives clear guidance on a choice of $\alpha$ in this initialisation scheme depending on whether you want exponential growth, shrinkage to zero, or approximately constant trajectory length through the network in expectation.  We will gladly include a brief explanation of these implications of our results and experiments in an updated draft.
>
> Moreover, outside direct application, these results give underpinning theory on the random object generated by a random network.  This is of interest from a purely mathematical point of view.  We expect this will also be helpful when considering random initialisation of GANs as a data model, where we show how one can control the complexity of the data model through the variance and sparsity.

---

> ### Author Response · Authors · 2019-11-08
> **Response to Review #2, Part 2**
>
> Our responses continue below:
>
> •	“However, it looks everything is done in a normal manner and nothing special happens”
>
> The proof is designed to be clear, easy to understand and generalise.  This proof is in no way an obvious extension of the proof of Raghu et al. which is substantially more complex than it need be, and this complexity is linked to dense Gaussian weight matrices.
>
> •	“What does the "restriction" of $\hat{w}_i$ mean?”
>
> As per our definition of the sparse random neural network, some entries of the weight matrix will be distributed according P, and others will be zero. The phrase “$\hat{w}_i$  is restriction of any row of a weight matrix to its P distributed entries” means that $\hat{w}_i$ is the vector you get when you take a row of the weight matrix delete all the zero entries, i.e. a vector containing only the entries which are P-distributed. We will add a suitable definition of the term ‘restriction’ to clarify this.
>
> •	“Does the condition $\mathbb{E}[] >= M \|u\|$ hold for any $u$ and $M$?”
>
> The condition on $\hat{w}_i$ must hold for some constant $M$ for any constant vector $u$. Thank you for pointing out the lack of clarity here, we will fix the phrasing to make this clear.
>
> •	“Finally, I have a simple question about the relationship between the scale of the output and the trajectory length. Let W be a weight and x be an input. When the smallest singular value of $W$ is $m$, $x$ is amplified at least by $m$, i.e., $\|Wx\| >= m\|x\|$ for any $x$. This means that (a part of) the trajectory growth is explained by the smallest singular value of the sparse random matrix $W$. Can you clarify the difference?”
>
> Thank you for raising this.  Indeed, one could lower bound the length as a product of (suitably modified) lower singular values of the weight matrices.  Such a bound would be a lower bound on the trajectory length rather a lower bound on the *expected* trajectory length.  While also very interesting, this is a different phenomenon and one which would have an even smaller lower bound. In particular, consider as an example $\sigma_w=1/\sqrt{k}$ and the ReLU setting to zero half of the pre-activation values.  In such a case the expected smallest singular value is $1-\sqrt{1/2}\approx 0.29$ (see [1]), in contrast with the value of $1/\sqrt{2\pi}\approx 0.4$ from $\alpha=1$ and $\sigma_w=1/\sqrt{k}$ in Corollary 1.  Moreover, for a rigorous bound, as in our case, one would need to justify that the expected lower singular with half the rows active would be appropriate (it gets much smaller as more rows become active), otherwise one would need to include further information about the distribution of rows and possibly the lower restricted isometry constants of the weight matrices.  Lastly, such bounds would be in terms of expected values of the lower singular values of random matrices, which are in general less precise for non-Gaussian distributions than the bounds we obtain which only require considering the bound involving $M$ in Theorem 2.
>
>
>
> [1] Rudelson, Mark, and Roman Vershynin. "Smallest singular value of a random rectangular matrix." Communications on Pure and Applied Mathematics: A Journal Issued by the Courant Institute of Mathematical Sciences 62.12 (2009): 1707-1739.

---

### Official Review · AnonReviewer1 · 2019-10-18
**Official Blind Review #1**

**Rating:** 6

**Review:**

This submission proposes an alternative way to lower bound the trajectory growth through random networks. It generalizes to a variety of weights distributions. For example, the authors showcase their approach on sparse-Gaussian, sparse-uniform, and sparse-discrete-valued random nets and prove that trajectory growth can be exponential in depth with these distributions, with the sparsity appearing in the base of the exponential.

I give an initial rating of weak accept because (1) the paper is well written and well-organized. (2) the numerical simulation results support the claims and proofs. (3) the investigation on sparsely connected networks seems timely. However, I'm not an expert in this area. It also seems that most derivation and insights are from previous literature Raghu 2017, which makes the contribution of this submission limited.

I have a question which may be invalid. For Figure 3, the observed expectation matches perfectly with the the lower bound for all three distributions. This seems amazing, have the authors try with other dataset or other settings to do this experiment? Did it always match perfectly?

**Experience Assessment:**

I do not know much about this area.

**Review Assessment: Checking Correctness Of Derivations And Theory:**

I assessed the sensibility of the derivations and theory.

**Review Assessment: Checking Correctness Of Experiments:**

I assessed the sensibility of the experiments.

**Review Assessment: Thoroughness In Paper Reading:**

I read the paper at least twice and used my best judgement in assessing the paper.

---

> ### Author Response · Authors · 2019-11-08
> **Response to Review #1**
>
> Thank you for your thorough reading of our paper, and for your positive feedback on the paper’s writing, results, and timeliness.
>
> Let us respond to specific points raised in your review.
>
> •	“I have a question which may be invalid. For Figure 3, the observed expectation matches perfectly with the lower bound for all three distributions. This seems amazing, have the authors try with other dataset or other settings to do this experiment? Did it always match perfectly?”
>
> The observed expectations (solid lines) in Figure 3 match perfectly across all three distributions, though they are all above their distributions' lower bounds (dashed lines). Yes, this is indeed remarkable, and yes, this is reproduced when choosing different datapoints, or indeed even points chosen uniformly at random, and connecting them with a straight line, or paths which are arcs in two or more dimensions. We think that this is a significant empirical observation, and one of the paper’s contributions, insofar as it indicates that there may be a universal dependence in expectation (within this class of distributions) on the standard deviation and sparsity level, such that the specific distribution is not important. We will edit this section in the paper to highlight that these variations of the experiments produce the same results, and include relevant extra figures in the appendices. Thank you for the suggestion. We are also considering some experiments on trained networks which might illustrate whether this is true more generally in over-parameterised networks.
>
> •	“It also seems that most derivation and insights are from previous literature Raghu 2017, which makes the contribution of this submission limited.”
>
> It is true that Raghu et al. deserve full credit for (1) the idea of considering expected lower bounds of trajectory growth and (2) doing so by considering the growth of a small piece of the trajectory, both of which inspired our work. However, a direct comparison of the proofs in our manuscript with those by Raghu et al. would show that it is not the case that “most derivation and insights are from Raghu et al.”. On the contrary, our result is a more general one, and our proof is fundamentally different. To be precise, the innovation of our proof is the strategy for explicitly accounting for the statistical dependence of an infinitesimal piece of the trajectory $dz$ on $z$ in such a way which allows us to prove a more general and more precise result. Conversely, the key fundamentals of their proof *limit* extension to more general cases: In particular, their proof is based on splitting the components of $dz$ into parts which are perpendicular and parallel to $z$ (which we do not do). One consequence of this strategy is that it requires that, after splitting the weight matrix into parallel and perpendicular parts $W_\top$ and $W_\|$, that these matrices are independent random matrices (their lemma 2). Everything from this point on in their proof relies on this fact – and *this is only true for Gaussian random matrices*. Thus, their proof strategy has no clear extension to sparse matrices and other distributions. Ours is not an extension *of their proof*, but rather is a completely different method to derive more general and more accurate bounds. A consequence of this is that in the specific standard Gaussian case, our work provides an alternative and more straightforward proof of a similar result to theirs. We apologise if in attempting to ‘give credit where credit is due’, we did not make clear enough in our writing the extent to which Raghu et al.’s work did not in any way constitute or provide a blueprint for the key innovation of this work, but rather was the inspiration and the first such bounds in the case of dense Gaussian weight matrices.

---

### Official Review · AnonReviewer4 · 2019-11-03
**Official Blind Review #4**

**Rating:** 3

**Review:**

Summary: The authors examine trajectory growth of deep ReLU neural networks whose weights come from a (random) “sparse-Gaussian”, “sparse-uniform”, and “sparse-discrete” distribution. They give definitions of these distributions in the paper. They do this by extending the proof of Raghu (2017) so that it can handle more general distributions than the standard Gaussian. They also provide some numerical experiments verifying their theories.

Strengths: The paper is well-written and the proofs are clearly explained. I’m grateful that the authors specifically mentioned where their proof deviates from the original and they clearly delineate how their proof method extends Raghu (2017)

Weaknesses: This is an interesting direction, but I do not believe there are enough results to constitute an accept. If the authors are following Raghu (2017), then I would have also liked to see analysis on trained networks as done in Raghu (2017) for example. I also think the title of the paper is too general for the specific results contained in the paper, namely sparsity should at least be mentioned in the title.

**Experience Assessment:**

I have read many papers in this area.

**Review Assessment: Checking Correctness Of Derivations And Theory:**

I assessed the sensibility of the derivations and theory.

**Review Assessment: Checking Correctness Of Experiments:**

I assessed the sensibility of the experiments.

**Review Assessment: Thoroughness In Paper Reading:**

I read the paper at least twice and used my best judgement in assessing the paper.

---

> ### Author Response · Authors · 2019-11-08
> **Response to Review #4, Part 1**
>
> Thank you for thoroughly reading our submission and your kind remarks regarding the writing and clarity of the proofs.
>
> Let us speak to the weaknesses listed:
>
> •	“I would have liked to see analysis on trained networks as done in Raghu (2017) for example.”
>
> The manuscript’s focus is on developing mathematically rigorous theoretical lower bounds on the expected length of a trajectory passed through a random deep network. The numerical experiments included are to illustrate the parameter dependencies of our bounds in the main Corollaries 1-3, and in doing so to show any gaps between the theory and results observed in practice.  Specifically, in Figure 2, to show the exponential dependence on depth given different sparsity parameters, in Figure 3(a) to illustrate the growth factor’s dependence on the weight variance and remarkable similarity across the different distributions (as noted too by Reviewer 3), in  Figure 3(b) to show the $\alpha$ dependence of the growth factor, again with remarkable similarity for different  distributions, and in Figure 4 to explain the qualitative difference between the form of the bound and the observed behaviour in the $\alpha$ dependence.  While we appreciate that Raghu et al. included some numerical experiments which used trained or partially trained networks, these experiments were to convey different phenomena discussed in the more wide reaching paper (for example, Figure 6 looks at the impact of noise in different layers of a trained network on accuracy; Figure 7 shows the accuracy of nets which have only one layer trained, for different choices of trained layer; Figure 8, 9, and 18 show that training without batch normalisation increases trajectory length, suggesting a potential role of batch normalisation and motivating their own regularisation method).  We have instead written a focused paper whose aim is to give a general framework by which the exponential expected length growth can be derived for large class of distributions (which includes classical Gaussian results of Raghu et al., but also include uniform and discrete distributions, and in all cases allow the weight matrices to be sparse).  Experiments on trained networks like those mentioned above would be very interesting, but would not shed any further light on our bounds or parameter relationships. More generally experiments on trained networks are complicated by there being many notions of what would constitute an ‘expectation’ over a trained network’s weights, and secondly, because the results would be conflated by the issue of how much of the trajectory which was passed through the network was within or between classes of the trained network (for example, we would not necessarily expect that portions of a trajectory *within a specific class* would grow exponentially through a trained network.) We are currently considering what experiments on trained networks might be most illuminating given these difficulties. One option is to try show whether the $\sigma_w$ dependence remains the same in different trained, over-parameterised networks; we aim to complete these and report back before the end of the review discussion period. We welcome suggestions from the reviewer.

---

> ### Author Response · Authors · 2019-11-08
> **Response to Review #4, Part 2**
>
> Our responses continue below:
>
> •	   The aforementioned critiques were listed after the statement “I do not believe there are enough results to constitute an accept.”
>
> Besides the issue of experiments on trained networks, we feel that this comment may be a consequence of framing our work as “extending the proof of Raghu et al. (2017)”. We apologise if we have incorrectly framed our contribution in this way. The impression we have created that we have simply extended their proof is misleading in the sense that it implies that once you have read and understood Raghu et al.’s proof, there is a natural extension to the more general case we consider. On the contrary, the key fundamentals of their proof *limit* extension to more general cases: In particular, their proof is based on splitting the components of an infinitesimal piece of the trajectory $dz$ into parts which are perpendicular and parallel to $z$ (which we do not do). One consequence of this strategy is that it requires that, after splitting the weight matrix into parallel and perpendicular parts $W_\top$ and $W_\|$, that these matrices are independent random matrices (their lemma 2). Everything from this point on in their proof relies on this fact – and *this is only true for Gaussian random matrices*. Thus, their proof strategy has no clear extension to sparse matrices and other distributions. Ours is not an extension *of their proof*, but rather is a completely different method to derive more general and more accurate bounds. A consequence of this is that in the specific standard Gaussian case, our work provides an alternative and more straightforward proof of a similar result to theirs.
>
> What we were attempting to credit Raghu et al. with is (1) the idea of considering expected lower bounds of trajectory growth and (2) doing so by considering the growth of a small piece of the trajectory, a starting point we share. Beyond this, however, our result is a more general one, and the proofs are fundamentally different, in particular in our explicit accounting for the conditional dependence of $dz$ on $z$, and everything which follows, which is the real key to the generality of our result. We encourage the reviewer to contrast the proofs in Raghu et al. (2017) with those in our manuscript.
>
> •	“I also think the title of the paper is too general for the specific results contained in the paper, namely sparsity should at least be mentioned in the paper.”
>
> Thank you for raising this.  We take your point that a number of very different papers could use the same title we began with.  In particular one could consider a more experimental investigation, and sparsity is not mentioned which, while the results do include the case of dense networks ($\alpha=1$), sparsity is one of our innovations that should be highlighted.  Taking this into account we propose changing the title to: “Trajectory growth lower bounds for random sparse deep ReLU networks”.  We choose this title to convey both that the results are theoretical lower bounds and to emphasise the sparsity.  Thank you.

---

### Author Response · Authors · 2019-11-09
**Revised submission uploaded**

We thank the reviewers again for their thorough reading of our paper and thoughtful comments.

We have uploaded a revised submission which makes the changes mentioned in our direct responses to each review. In particular:

•	We changed the title to “Trajectory growth lower bounds for random sparse deep ReLU networks”
•	We clarified the condition involving $M$ and $u$ in Theorem 2, and shifted to Section 1.1 (top of page 3) the definition of the restriction of a row of the weight matrix to its P-distributed entries.
•	We have highlighted the fact that the lower bounds give some principled guidance on the choice of the combination of $\sigma_w$ and $\alpha$ (on page 4, below Corollary 3) depending on the growth properties you want for the network at initialisation. We also highlight (on page 9, end of paragraph 2) how the numerical experiments give similar guidance on these parameter choices which may accord more precisely with what occurs in practice, as compared with the theoretical lower bounds.
•	We have highlighted that the universal dependence of the growth factor on $\sigma_w$ and $\alpha$ across distributions (shown in Figures 3a and 3b) remain true if you repeat the experiments with different trajectories and different, random datapoints (page 9, paragraph 2). We have included Figures for these experiments in Figure 5, Appendix C.

The proposed experiments on trained networks are still in the works and as of yet are not included in the revised submission.

---

### Author Response · Authors · 2019-11-15
**Final revision uploaded - includes experiments with trained networks**

We have now uploaded a further revised submission with an extra two figures in Appendix C.2 which show the results of some experiments on trained networks, and a pointer to these figures at the end of Section 4 on page  9.

The results indicate, firstly, that even in trained networks, trajectory growth through the network can still be exponential, and secondly, that the expected growth factor still depends on the standard deviation of the trained weights in an approximately linear relationship - both of which resemble the results on random nets. It also seems that the growth factor from one layer to the next tends to be somewhat larger for trajectories connecting points which belong to different classes, than trajectories connecting points from the same class. However, in contrast to the results for random networks,  expected trajectory growth through trained networks appears not to be  trajectory independent.

---

### Decision · Program_Chairs · 2019-12-19

**Decision:**

Reject

**Comment:**

This article studies the length of one-dimensional trajectories as they are mapped through the layers of a ReLU network, simplifying proof methods and generalising previous results on networks with random weights to cover different classes of weight distributions including sparse ones. It is observed that the behaviour is similar for different distributions, suggesting a type of universality. The reviewers found that the paper is well written and appreciated the clear description of the places where the proofs deviate from previous works. However, they found that the results, although adding interesting observations in the sparse setting, are qualitatively very close to previous works and possibly not substantial enough for publication in ICLR. The revision includes some experiments with trained networks and updates the title to better reflect the contribution. However, the reviewers did not find this convincing enough. The article would benefit from a deeper theory clarifying the observations that have been made so far, and more extensive experiments connecting to practice.